# Sleep Bruxism Episodes in Patients with Obstructive Sleep Apnea Syndrome Determined by In-Laboratory Polysomnography

**Dong Hyun Kim [1] , Sang Hwa Lee [2],* and Sang Haak Lee [3]**

[1]   Department of Otorhinolaryngology-Head and Neck Surgery, Incheon St. Mary's Hospital,
     College of Medicine, The Catholic University of Korea, Seoul 06591, Korea; monolithkim@catholic.ac.kr
[2]   Department of Dentistry, Eunpyeong St. Mary's Hospital, College of Medicine,
     The Catholic University of Korea, Seoul 06591, Korea
[3]   Department of Internal Medicine, Division of Pulmonary, Critical Care and Sleep Medicine,
     Eunpyeong St. Mary's Hospital, College of Medicine, The Catholic University of Korea, Seoul 06591, Korea;
     mdlee@catholic.ac.kr
*    Correspondence: justina@catholic.ac.kr; Tel.: +82-2-2030-2828

**Abstract:** Findings on sleep bruxism (SB) in patients with obstructive sleep apnea syndrome (OSAS) are controversial, and some of these findings have relied on in-laboratory polysomnography (PSG). We aimed to identify the factors associated with SB episodes in 100 patients with OSAS using in-laboratory PSG records. Subjects with OSAS were divided into those with and without SB episodes. We analyzed the differences in patient characteristics and PSG indices. Age, gender, height, weight, body mass index, neck, waist, and hip circumferences, and the rates of hypertension and diabetes mellitus were not significantly different between the two groups. A greater proportion of stage N2 sleep in the total sleep time, longer total sleep time, longer sleep time in a supine position, shorter sleep time in a nonsupine position, lower apnea–hypopnea index (AHI), lower AHI regardless of sleeping position, lower AHI during nonrapid eye movement sleep, and higher mean oxygen saturation level were associated with SB episodes in patients with OSAS. Among these factors, longer sleep time in a supine position remained a statistically significant factor in multivariate analysis. We conclude that longer sleep time in a supine position (especially >280 min) might be associated with SB episodes in patients with OSAS.

**Keywords:** obstructive; polysomnography; sleep apnea; sleep bruxism

## 1. Introduction

Obstructive sleep apnea syndrome (OSAS) is a prevalent sleep disorder characterized by repeated upper airway collapse. It can cause various problems, including daytime sleepiness, neurocognitive disorders, cardiovascular and metabolic disorders, and even traffic accidents [1,2]. Sleep bruxism (SB) is a masticatory muscle activity during sleep that is characterized as rhythmic (phasic) or nonrhythmic (tonic) and is not a movement disorder or a sleep disorder in otherwise healthy individuals [3]. SB is sometimes seen in patients with OSAS. Although continuous positive airway pressure (CPAP) therapy is highly effective for treating patients with OSAS, there are several confounding issues of CPAP therapy, such as problems with adaptation to devices and compliance [4]. An oral appliance is one superior alternative approach to CPAP for patients suffering from OSAS [5,6]. SB in a patient with OSAS might affect the treatment using an oral appliance instead of CPAP therapy. Considering that oral appliances have better preference and compliance than CPAP [7,8] and that well-designed oral appliances can control both SB and OSAS simultaneously [9,10], these might be considered more useful

than CPAP when the patient has SB during OSAS, especially for mild and moderate OSAS. However, the relationship between OSAS and SB is unclear. The criteria used to investigate the relationship between SB and OSAS have varied between studies and the results are controversial [11–16]. Therefore, further studies on SB in patients with OSAS are necessary.

In-laboratory polysomnography (PSG) is the gold standard for diagnosing OSAS [17] and detecting SB episodes [18]. However, this technique is time-consuming and expensive and has long waiting periods. Few studies have examined the association between SB and OSAS using in-laboratory PSG. Also, there has been insufficient research on the association between SB episodes identified by in-laboratory PSG and self-reported SB. Here we aimed to investigate patients' characteristics in cases of OSAS with SB episodes and related PSG indices using in-laboratory PSG. For this, we compared subjects with and without SB episodes among patients with OSAS rather than healthy controls because we wished to consider the influences between factors that might affect the combination of OSAS and SB. We evaluated whether a single overnight in-laboratory PSG session would be suitable for assessing SB episodes in patients with OSAS.

## 2. Materials and Methods

### 2.1. Study Design and Study Population

We performed a retrospective chart review for all patients who visited our hospital for snoring, sleep apnea, or drowsiness during the day and underwent in-laboratory full-night PSG between March 2017 and February 2019. Hypertension and diabetes mellitus were determined based on patients' records at hospital visits, considering factors such as taking drugs prescribed for these conditions. Self-reported SB was determined based on questionnaires as to whether the patients had been told or noticed by themselves that they grind their teeth or clench their jaws when they are asleep. Inclusion criteria were as follows: Age ≥18 years, and an apnea–hypopnea index (AHI) ≥ 5 based on in-laboratory PSG. Patient data were excluded if:

1. they were aged <18 years;
2. AHI was <5;
3. they used any medications known to affect sleep or breathing;
4. In-laboratory PSG was performed for a therapeutic purpose, such as the titration of CPAP; or
5. the patient had neurological disorders, epilepsy, neuromuscular disease, a history of upper airway surgery, or any severe mental illness.

Of the 379 subjects who underwent in-laboratory PSG, the data for 100 who met the above criteria were analyzed. The data were divided into groups of patients with and without SB episodes (Figure 1).

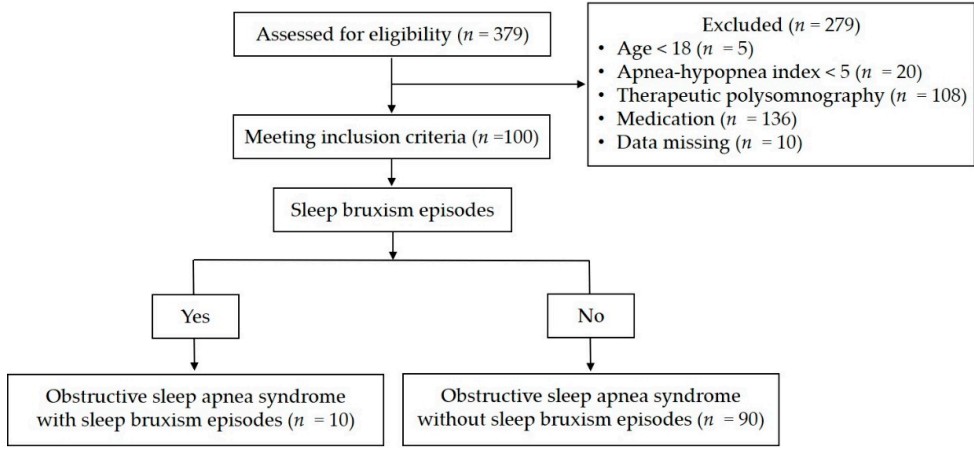

**Figure 1.** Study flow diagram.

## 2.2. Ethics Statement

The Institutional Review Board of the Eunpyeong St. Mary's Hospital approved this study (IRB No. PC18RESI0011).

## 2.3. In-Laboratory PSG

Sleep studies were performed using a standard polysomnograph (SomnoStar Pro®; VIASYS Healthcare Inc., Yorba Linda, CA, USA). Qualified polysomnographic sleep technicians and physicians completed the testing and scoring using the American Academy of Sleep Medicine (AASM) guidelines [19,20]. We recorded the results for electroencephalography (EEG; frontal, central, and occipital regions), electrooculography, electromyography (EMG; chin, masseter muscle, and right/left tibial regions). Electrocardiograms and a pressure cannula, a thermistor, respiratory inductance plethysmography, and pulse oximetry were used. Snoring was measured using a microphone attached near the thyroid cartilage. A video camera (EB-3645N; Dongyang Unitech Co., Seoul, S. Korea) was installed 1 m above the patient's bed near the ceiling. Polysomnograms were assessed in 30-s epochs following the AASM standard criteria for sleep scoring. The following PSG outcome variables were included in the analysis: total sleep time; sleep onset latency; sleep efficiency; the percentages of stages N1, N2, N3, and rapid eye movement (REM) sleep; apnea index (AI); AHI; AHI during nonrapid eye movement (NREM) sleep; the AHI of REM sleep; the sleep time in the nonsupine position; the sleep time in the supine position; the AHI in the nonsupine position; the AHI in the supine position; mean oxygen saturation; lowest oxygen saturation; mean oxygen desaturation; and the sound intensity of snoring. Apnea was defined as a drop in the peak signal excursion by $\geq 90\%$ of the pre-event baseline using a thermistor or as complete airflow cessation for at least 10 s. Hypopnea was defined as a peak signal excursion drop by $\geq 30\%$ of the pre-event baseline using a pressure cannula and a $\geq 3\%$ oxygen desaturation from the pre-event baseline, whether or not it was associated with arousal. AHI was defined as the total number of apnea and hypopnea events within 1 h of sleep.

## 2.4. SB Episodes

SB episodes were diagnosed with audio/video recordings and EMGs of the chin and masseter muscle during PSG. Each SB episode was scored according to AASM criteria [19,20], which included an EMG activity more than twice the amplitude of the background EMG and an EMG burst that was not separated by more than 3 s within the same episode [21]. A constant burst episode lasting longer than 2 s or three or more bursts that were 0.25–2 s long was considered to be a SB episode. Simultaneously, tooth grinding sounds and typical jaw movements were required to consider these EMG activities or bursts as SB episodes. EMG activities or bursts associated with sounds or behaviors involved in swallowing, coughing, and face rubbing were not considered as SB episodes. Changes in EMG activity or bursts that occurred when the patient's face was not visible, such as when their face was hidden by a blanket, were not considered as a SB episode.

## 2.5. Statistical Analysis

Statistical analysis was performed using MedCalc for Windows (version 19.4.1; MedCalc Software, Mariakerke, Belgium). Descriptive statistics were used to evaluate the influence of clinical parameters. Quantitative data are presented as the mean value (standard deviation). Normality of data distribution was verified using the Kolmogorov–Smirnov test. The $\chi^2$ test and Fisher's exact test were applied to evaluate the difference in clinical characteristics between patients with OSAS with and without SB episodes. The Mann–Whitney nonparametric *U*-test or Student's *t* test were used in analyzing overall differences between the two groups. Multivariate analysis using linear regression was performed on factors that showed significance in the univariate analysis. A *p* value of <0.05 was considered statistically significant.

## 3. Results

*3.1. General Characteristics in Patients with OSAS with and without SB Episodes*

The features of the 100 subjects with and without SB episodes in this study are summarized in Table 1. The age range in those with SB episodes was 20–73 years, while it was 18–76 years in those without SB episodes. Age, gender, height, weight, body mass index, neck circumference, waist circumference, hip circumference, and the prevalence of self-reported SB, hypertension, or diabetes mellitus were not significantly different between the groups (Table 1). Among the 100 patients with OSAS, 10 had SB episodes (95% confidence interval [CI], 4.0–16.0%) measured by in-laboratory PSG. Among these patients, 16 subjects had self-reported SB (95% CI, 8.7–23.3%) and three of these had SB episodes (18.8%; 95% CI, 2.1–64.6%) identified by in-laboratory PSG. Of 84 patients without self-reported SB, seven had SB episodes (8.3%; 95% CI, 1.5–7.9%) identified by in-laboratory PSG. Although there were more SB episodes in the patients with self-reported SB than in those without it, this was not statistically significant.

**Table 1.** Clinical characteristics in patients with obstructive sleep apnea syndrome (OSAS) with and without sleep bruxism (SB) episodes.

| Demographics | With SB Episodes ($n = 10$) | Without SB Episodes ($n = 90$) | $p$ |
|---|---|---|---|
| Age (years) | 43.4 (16.1) | 48.5 (13.9) | 0.29 |
| Gender (male/female) | 10/0 | 80/10 | 0.59 |
| Height (cm) | 173.8 (9.0) | 169.5 (7.7) | 0.11 |
| Weight (kg) | 82.2 (12.5) | 80.5 (19.3) | 0.34 |
| BMI (kg/m$^2$) | 27.3 (4.3) | 27.9 (5.7) | 0.91 |
| Neck circumstance (cm) | 40.1 (3.2) | 39.6 (3.9) | 0.68 |
| Waist circumstance (cm) | 98.3 (11.0) | 99.3 (13.7) | 0.90 |
| Hip circumstance (cm) | 103.3 (8.2) | 103.3 (10.5) | 0.84 |
| Self-reported SB (−/+) | 7/3 | 77/13 | 0.20 |
| Hypertension (−/+) | 6/4 | 54/36 | 1.00 |
| Diabetes mellitus (−/+) | 9/1 | 76/14 | 1.00 |

Results are presented as the mean value (standard deviation). OSAS, obstructive sleep apnea syndrome; SB, sleep bruxism; BMI, body mass index. Self-reported SB (±) means whether the patients had been told or noticed by themselves that they grind their teeth or clench their jaws when they are asleep. Hypertension (±) or diabetes mellitus (±) means that the patient was not or was diagnosed and taking medications prescribed for these conditions, respectively.

*3.2. Comparative Analysis of PSG Indices in Patients with OSAS with and without SB Episodes*

There were significant differences in the following factors between the patients with and without SB episodes: total sleep time, the percentage of stage N2 sleep in the total sleep time, sleep time in a nonsupine or supine position, AHI, the AHI of NREM sleep, the AHI in a nonsupine position, the AHI in a supine position, and mean oxygen saturation (Table 2). The total sleep time in patients with SB episodes was longer than in those without. Stage N2 sleep in patients with SB episodes occupied a larger percentage of the total sleep time than in those without. Sleep time in the supine position in patients with SB episodes was longer than in those without. In contrast, sleep time in the nonsupine position in patients with SB episodes was shorter than in those without. The AHI in patients with SB episodes, regardless of sleep position, was lower than in those without SB episodes. The AHI during NREM sleep in patients with SB episodes was lower than in those without. The mean oxygen saturation level in patients with SB episodes was higher than in those without. Of these significant PSG indices, sleep time in the supine position remained a statistically significant variable in the multivariate analysis (Table 3). The cut-off point for sleep time in the supine position was determined based on a receiver operating characteristic curve (Figure 2). According to this analysis, the cut-off point was set at a sleep time in the supine position of 280 min, which indicated SB episodes with a sensitivity and specificity of 80.0% (95% CI, 44.4–97.5%) and 70.0% (95% CI, 59.4–79.2%), respectively, giving a predictive accuracy of 0.745 (95% CI, 0.648–0.827; $p < 0.01$). The positive and negative likelihood ratios were 2.67 and 0.29, respectively. Sleep efficiency, sleep onset latency, the percentages of stages N1, N3,

or REM sleep in the total sleep time, AI, AHI during REM sleep, the lowest oxygen saturation level, the mean oxygen desaturation level, and the sound intensity of snoring were not significantly different between the patients with and without SB episodes (Table 2).

**Table 2.** Univariate statistical analysis of polysomnographic indices in 100 patients with OSAS with and without SB episodes.

| Index | With SB Episodes (n = 10) | Without SB Episodes (n = 90) | p |
|---|---|---|---|
| TST (min) | 363.2 (72.7) | 331.6 (56.0) | 0.03 |
| SE (%) | 86.2 (14.1) | 81.5 (14.7) | 0.14 |
| SL (min) | 3.5 (2.3) | 6.4 (6.5) | 0.25 |
| Stage N1 sleep (% of TST) | 24.2 (13.3) | 34.3 (16.5) | 0.06 |
| Stage N2 sleep (% of TST) | 50.0 (10.6) | 41.4 (11.5) | 0.03 |
| Stage N3 sleep (% of TST) | 9.3 (7.0) | 7.7 (7.7) | 0.33 |
| Stage REM sleep (% of TST) | 16.5 (6.7) | 16.6 (6.9) | 0.96 |
| ST in the nonsupine position (min) | 46.9 (71.5) | 104.6 (86.9) | 0.04 |
| ST in the supine position (min) | 316.4 (92.8) | 227.2 (97.0) | 0.01 |
| AI (events/h) | 6.9 (9.3) | 21.0 (26.3) | 0.13 |
| AHI (events/h) | 25.5 (16.6) | 45.1 (28.2) | 0.03 |
| AHI of NREM sleep (events/h) | 23.4 (18.1) | 44.7 (30.2) | 0.02 |
| AHI of REM sleep (events/h) | 34.6 (19.8) | 45.2 (25.4) | 0.25 |
| AHI in the nonsupine position(events/h) | 7.6 (17.2) | 23.1 (30.3) | 0.02 |
| AHI in the supine position (events/h) | 28.3 (17.8) | 56.0 (30.7) | 0.01 |
| Mean oxygen saturation (%) | 94.8 (2.0) | 91.6 (5.1) | 0.03 |
| Lowest oxygen saturation (%) | 83.8 (8.5) | 80.4 (9.6) | 0.26 |
| Mean oxygen desaturation (%) | 14.5 (8.5) | 17.6 (9.5) | 0.28 |
| Sound intensity of snoring (dB) | 59.9 (9.1) | 60.3 (14.0) | 0.69 |

Results are presented as the mean value (standard deviation). OSAS, obstructive sleep apnea syndrome; SB, sleep bruxism; TST, total sleep time; SE, sleep efficiency; SL, sleep onset latency; NREM, nonrapid-eye movement; ST, sleep time; AI, apnea index; AHI, apnea–hypopnea index; REM, rapid eye movement.

**Table 3.** Multiple linear regression analysis of significant factors in univariate analysis.

| Index | Coefficient (β) | SE | t | p |
|---|---|---|---|---|
| Stage N2 sleep (% of TST) | 0.003 | 0.003 | 1.080 | 0.28 |
| ST in the supine position (min) | 0.001 | 0.001 | 2.871 | 0.01 |
| AHI (events/h) | −0.001 | 0.002 | −0.471 | 0.64 |
| Mean oxygen saturation (%) | 0.006 | 0.010 | 0.574 | 0.57 |

$R^2 = 0.137$; adjusted $R^2 = 0.101$; SE, standard error; TST, total sleep time; ST, sleep time; AHI, apnea–hypopnea index.

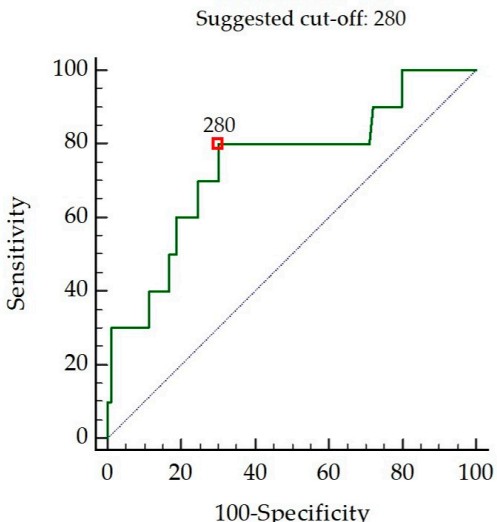

**Figure 2.** Receiver operating characteristic curve (ROC) suggesting the optimal sleep time in the supine position as a cut-off point in minutes for indicating its suitability to recognize SB episodes.

## 4. Discussion

In this study, we found that sleep stage, sleep position, AHI, and oxygen saturation might be more strongly associated with SB episodes than the patients' characteristics, and that longer sleep time in the supine position might be more significantly associated with SB episodes in patients with OSAS. The AHI in patients without SB episodes was higher than in such patients without SB episodes in both the supine and nonsupine positions during sleep in this study. We found that low AHI was more likely to be linked to SB episodes in OSAS, similar to previous reports [11,14]. One of the hypotheses relating to SB episodes is that they might protect against OSAS by protruding the mandible and restoring airway patency [14,22]. However, this mechanism may not be adequate to prevent the airway from collapsing in cases of severe OSAS [14]. Higher oxygen saturation in the study patients was more likely to cause SB episodes in this study. We postulate that when the AHI is high or when the oxygen saturation level is low in patients with OSAS, the mechanisms involved in opening the mouth to breathe are facilitated preferentially over those used to close the mouth for SB episodes, to resolve or compensate for apnea or hypopnea, and to improve oxygen supply. Other mechanisms might be involved in a high AHI, such as excessive respiratory efforts and increased respiratory rate [14]. Therefore, we suggest that a low AHI might induce SB episodes to restore the airway and conversely that a high AHI might reduce or prevent them. However, further studies are needed to verify our hypothesis and determine how high an AHI is associated with reducing or preventing SB episodes. The total sleep time and sleep time in the supine position were longer in these patients with SB episodes than in those without them. The longer the total sleep time, the more opportunities there are for SB episodes to occur. The association of sleep time in the supine position with SB episodes is consistent with a previous report [23]. Here, we found that a sleep time in the supine position of >280 min might be associated with SB episodes in these patients. Differences in occlusal contact patterns of teeth according to sleeping position might also be associated with SB episodes [24,25]. Our study did not include the subjects' dental conditions; therefore, it was impossible to investigate the association between SB episodes and the occlusal contact pattern of teeth according to sleeping positions. Well-designed and prospective studies examining tooth occlusal contact patterns will be needed to explore such possible associations. SB episodes can occur during any sleep stage, such as in NREM-like stages N1 and N2 and during REM sleep, or during arousal [26,27]. In this study, a more prolonged N2 sleep phase was linked to more SB episodes. Some general population studies have reported that SB decreases with age [28–30]. There are also reports that SB is not associated with gender in adults [26,31], although SB seems to be more prevalent in men [32]. Martynowicz et al. suggested that diabetes mellitus and male gender might be associated with SB in cases of OSAS [14]. In our study, however, SB episodes in patients with OSAS were not associated with the patients' characteristics, including age, gender, or the presence of hypertension or diabetes mellitus. However, genders were not equally represented in this study. Previous clinic-based and community-based studies have consistently reported a higher prevalence of OSAS in men than in women [33,34]. Because we compared cases with and without SB episodes among patients with OSAS rather than in healthy controls, this explains why there were more men than women in the composition of patients. However, we found that SB episodes were not associated with gender. Therefore, we postulate that the small number of female subjects in this study was associated with the effect of OSAS rather than with SB episodes. We also found no significant difference in snoring sound intensity between the patients with and without SB episodes. Therefore, there might be little or no association between SB episodes and snoring in such patients, as reported previously [28]. However, studies including frequency analysis of snoring sounds and snoring times are necessary to evaluate this possible association.

Here we observed inconsistencies between EMG activities and audio/video recordings. In some cases, only EMG activities were found, but no sound or typical jaw movements were observed in the audio/video recordings. These EMG activities might simply reflect rhythmic masticatory muscle activity. We suggest that longitudinal recordings are required along with EMG studies for documenting SB episodes because SB episodes show night-to-night variability [35]; thus, SB episodes cannot

necessarily be recorded on the day of testing. In this study, 16 patients with OSAS had self-reported SB. Only three (18.8%) among these patients had SB episodes over 1 night of in-laboratory PSG. This did not identify significantly more SB episodes in patients with than in those without self-reported SB ($p$ = 0.20). Further research is needed on how many PSG tests are necessary because repeated or multiple-night in-laboratory PSGs are very costly. There are even more restrictions on repeated or multiple-night in-laboratory PSGs in the current pandemic era of COVID-19. Therefore, portable PSGs might be considered an alternative method for repeated or multiple-night tests. Easy and comfortable approaches for audio/video recordings, such as smartphone applications [36], along with the use of portable PSG [37,38], should be developed.

This study had several limitations. We assessed self-reported SB, EMG activity, tooth grinding sounds, and typical jaw movements during sleep for SB episodes in these patients with OSAS. However, clinical signs such as dental attrition, abfraction, masseter muscle hypertrophy, and temporary morning jaw muscle pain or fatigue were not included for the evaluation. Because SB episodes can appear temporarily even in normal subjects, we recommend further studies, including clinical examination of the nature of SB [3]. Moreover, this was a retrospective study in a single institution and the number of patients with SB episodes was small ($n$ = 10). There was a great inequity in the number of patients between groups with and without SB episodes. Therefore, a prospective and well-controlled multicenter study involving more patients will be needed to confirm or expand our results. As our study was based on single-night, in-laboratory PSG results, the first-night effect associated with adaptation to the sleep laboratory environment should be considered when interpreting these results [35,39].

## 5. Conclusions

We found that a greater proportion of stage N2 sleep, longer sleep time in a supine position, lower AHI, and higher oxygen saturation were associated with SB episodes in these patients with OSAS. Longer sleep time in a supine position (especially over 280 min) might be more strongly associated with SB episodes in such patients.

**Author Contributions:** Conceptualization, D.H.K., S.H.L. (Sang Hwa Lee), and S.H.L. (Sang Haak Lee); Data curation, D.H.K.; Formal analysis, D.H.K.; Supervision, S.H.L. (Sang Hwa Lee) and S.H.L. (Sang Haak Lee); Writing—original draft, D.H.K.; Writing—review & editing, S.H.L. (Sang Hwa Lee) and S.H.L. (Sang Haak Lee). All authors have read and agreed to the published version of the manuscript.

**Funding:** This research received no external funding.

**Acknowledgments:** The authors thank Yun Jung Yang for help with the statistical analysis. The authors would also like to thank Soo Hyun Kim and Hyun Sun Ryu for their help in organizing the data.

**Conflicts of Interest:** The authors declare no conflict of interest.

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
