# Peer review of "Sleep Bruxism Episodes in Patients with Obstructive Sleep Apnea Syndrome Determined by In-Laboratory Polysomnography"

_applsci, doi:10.3390/app10238587_

Round 1
Reviewer 1 Report
I reviewed the manuscript entitled: " Sleep bruxism episodes in patients with obstructive sleep apnea syndrome determined by in-laboratory polysomnography” and found it relevant to the study of the controversial issue of the relation between sleep bruxism , and as such of interest to the readers of the Journal of Applied Sciences .
Some minor points of criticisms should be addressed before the manuscript can be considered for publication.
My comments are the following:
- General comment: Please avoid from repeating the same sentences in the introduction and the discussion.
- The definition of sleep bruxism according to reference 3 is:" Sleep bruxism is a masticatory muscle activity during sleep that is characterized as rhythmic (phasic) or non-rhythmic (tonic) and is not a movement disorder or a sleep disorder" and not as written in the introduction (lines 34-35).
- Please explain in more details the statement: "If the patient has SB during OSAS, this might affect the choice of oral appliance treatment 162 [4-6]" (lines 38-39 & 162-163)
- Please give references for the following statements:
- "An oral appliance is one superior alternative approach to CPAP for patients suffering from OSAS" (Lines 161-162).
- "Differences in occlusal contact patterns of teeth according to the sleeping position might also be related to SB episodes." (Lines 189-190).
Author Response
Thank you for your constructive comments.
Point 1: Please avoid from repeating the same sentences in the introduction and the discussion.
Response 1: Thank you for the valuable comment on the reader’s understanding of the introduction and discussion section. Based on the reviewer’s first comment, we rearranged the sentences from the discussion to the introduction part and avoided the same sentences in the introduction and the discussion. And we modified our introduction and discussion section to help the reader’s understanding.
Point 2: The definition of sleep bruxism according to reference 3 is:" Sleep bruxism is a masticatory muscle activity during sleep that is characterized as rhythmic (phasic) or non-rhythmic (tonic) and is not a movement disorder or a sleep disorder" and not as written in the introduction (lines 34-35).
Response 2: Thank you for your thoughtful comment about the corrections of our manuscripts. We modified the definition of sleep bruxism according to reference 3.
Point 3. Please explain in more details the statement: "If the patient has SB during OSAS, this might affect the choice of oral appliance treatment 162 [4-6]" (lines 38-39 & 162-163)
Response 3: According to your comment, we added more details to the statement below.
“Although continuous positive airway pressure (CPAP) therapy is highly effective for treating patients with OSAS, there are several confounding issues of CPAP therapy, such as problems with adaptation to devices and compliance [4]. An oral appliance is one superior alternative approach to CPAP for patients suffering from OSAS [5, 6]. SB in a patient with OSAS might affect the treatment using an oral appliance instead of CPAP therapy. Considering that oral appliances have better preference and compliance than CPAP [7, 8] and that well-designed oral appliances can control both SB and OSAS simultaneously [9, 10], these might be considered more useful than CPAP when the patient has SB during OSAS, especially for mild and moderate OSAS.”
Point 4: Please give references for the following statements:
"An oral appliance is one superior alternative approach to CPAP for patients suffering from OSAS" (Lines 161-162).
"Differences in occlusal contact patterns of teeth according to the sleeping position might also be related to SB episodes." (Lines 189-190).
Response 4: According to your comment, we added the references in our manuscript as below.
"An oral appliance is one superior alternative approach to CPAP for patients suffering from OSAS" (added references 5 and 6).
- Dieltjens M, Vanderveken O. Oral Appliances in Obstructive Sleep Apnea. Healthcare (Basel). 2019;7(4).
- Ng JH, Yow M. Oral Appliances in the Management of Obstructive Sleep Apnea. Sleep Med Clin. 2019;14(1):109-18.
"Differences in occlusal contact patterns of teeth according to the sleeping position might also be related to SB episodes." (added references 24 and 25).
- Phillips BA, Okeson J, Paesani D, Gilmore R. Effect of sleep position on sleep apnea and parafunctional activity. Chest. 1986;90(3):424-9.
25. Tago C, Aoki S, Sato S. Status of occlusal contact during sleep bruxism in patients who visited dental clinics - A study using a Bruxchecker(R). Cranio. 2018;36(3):167-73

Reviewer 2 Report
Thank you for your manuscript. The manuscript described using in-laboratory polysomnography to evaluate sleep bruxism (SB) episodes in patients with obstructive sleep apnea syndrome (OSAS) and aimed at finding the association between these conditions SB and OSAS. This retrospective study seems interesting, but the discussion and the attempt to find causation between variables is poor. Authors should elaborate more on which came first: the chicken or the egg.
Abstract:
I would suggest to underline the significant differences between study groups, and point out how they were different.
What it “sleep positional time”? and “sleep positional AHI”?
“N2 sleep” as well as “N1, N3” need explanation
Introduction must be revised and enhanced.
Please, revise lines 34-36 as according to https://pubmed.ncbi.nlm.nih.gov/29926505/, the paper that was cited by the authors of the current study:
“Sleep bruxism is a masticatory muscle activity during sleep that is characterised as rhythmic (phasic) or non-rhythmic (tonic) and is not a movement disorder or a sleep disorder in other-wise healthy individuals.”
Also, revise lines 40-43
Multiple times (line 56, 90, 91,92) there is such a strange sign present. Please, correct it.
The patients age range is not given.
The study groups: there is a great inequity in number of participants between groups with and without SB episodes. Also, Table 1 shows that females were hugely underrepresented in the study (10 females out of 100 patients). Is OSA syndrome more common among men? Is bruxism ”gender-related”? How all these issues could influence the results of the study? Please, explain and discuss.
Table 1: explain the following:
Are these the mean values presented in Table 1 and Table 2? If so, provide such info e.g. Mean (SD).
“History of SB” meaning?
“High blood pressure” define and explain whether patients (+) were taking medications or not? Same with DM.
Terms such as “supine positional sleep time” or “nonsupine positional sleep” should be verified and corrected.
Lines 133-134: revise giving more info on the differences between the results obtained for patients with OSAS, with and without SB episodes. Move here the info from lines 170-178 (from Discussion), underline the significant differences.
Line 158: please add “… in patients with OSAS” at the end of the sentence.
Line 165: revise “…assess the association”
Line 166: revise, “therefore” twice in one line
Discussion is poorly written. First part consists of either the repetition of statements from Introduction or statements (lines 170-178) that rather belong to the Results section not Discussion.
Line 178-180: the statement needs revision as to the causation: low AHI causes SB episodes or SB episodes cause low AHI? In patients with obstructive sleep apnea without SB episodes, severe apnea (high AHI) was observed in both supine and non-supine position during sleep and, in both sleeping positions, AHI was higher than in patients with SB episodes. Please discuss it further.
Line 186: not entirely true as most SB episodes occur in the stages N1 and N2 of sleep, called non-rapid eye movement (non-REM) sleep, what authors mentioned in lines 194-195.
Lines 198-199: revise the statement as in this study the age of participants is not given nor genders are equally represented.
Line 210: again define “patients had history of SB”
Line 217: there are portable devices to measure SB:
https://pubmed.ncbi.nlm.nih.gov/31261634/
https://pubmed.ncbi.nlm.nih.gov/25040303/
Author Response
Thank you for your constructive comments.
Point 1: Abstract: I would suggest to underline the significant differences between study groups, and point out how they were different.
Response 1: We appreciate your significant comment on the sentences in the abstract section. Based on the reviewer’s first comment, we underlined the considerable differences between the study group and point out how they were different as below.
“A greater proportion of stage N2 sleep in the total sleep time, longer total sleep time, longer sleep time in a supine position, shorter sleep time in a nonsupine position, lower apnea–hypopnea index (AHI), lower AHI regardless of sleeping position, lower AHI during nonrapid eye movement sleep, and higher mean oxygen saturation level were associated with SB episodes in patients with OSAS. Among these factors, longer sleep time in a supine position remained a statistically significant factor in multivariate analysis.”
Point 2: What it “sleep positional time”? and “sleep positional AHI”? “N2 sleep” as well as “N1, N3” need explanation
|
Response 2: Sleep positional time refers to the sleep time of supine or non-supine position. The expression has been modified to the sleep time of supine position or non-supine position for better understanding. Also, sleep positional AHI refers to the AHI of non-supine position or the AHI of the supine position. N2 sleep has modified the percentage of stage N2 in total sleep time. N1 has changed the percentage of stage N1 sleep in total sleep time. N3 has adjusted the percentage of stage N3 sleep in total sleep time. Point 3: Introduction must be revised and enhanced. Response 3: Thank you for the valuable comment on the reader’s understanding of the introduction section. We modified our introduction section to help the reader’s experience. |
|
Point 4: Please, revise lines 34-36 as according to Https://pubmed.ncbi.nlm.nih.gov/29926505/, the paper that was cited by the authors of the current study: “Sleep bruxism is a masticatory muscle activity during sleep that is characterised as rhythmic (phasic) or non-rhythmic (tonic) and is not a movement disorder or a sleep disorder in other-wise healthy individuals.” Response 4: Thank you for your thoughtful comment about the corrections of our manuscripts. We modified the definition of sleep bruxism according to reference 3. |
Point 5: Also, revise lines 40-43
Response 5: According to your comment, we revised the lines as below.
“In-laboratory polysomnography (PSG) is the gold standard for the diagnosis of OSAS and its severity. In-laboratory PSG recordings using chin–masseter electromyograms (EMGs) and audio–video recordings are regarded as the gold standard for detecting episodes of SB.
☞” In-laboratory polysomnography (PSG) is the gold standard for diagnosing OSAS [17] and detecting SB episodes [18].”
Point 6: Multiple times (line 56, 90, 91,92) there is such a strange sign present. Please, correct it.
Response 6: Thank you for your thoughtful comment about the corrections of our manuscripts. We tried to find and correct typos and strange signs.
Point 7: The patients age range is not given.
Response 7: We added the patient age range in the result section as below.
“The age range in those with SB episodes was 20–73 years, while it was 18–76 years in those without SB episodes.”
Point 8: The study groups: there is a great inequity in number of participants between groups with and without SB episodes. Also, Table 1 shows that females were hugely underrepresented in the study (10 females out of 100 patients). Is OSA syndrome more common among men? Is bruxism ”gender-related”? How all these issues could influence the results of the study? Please, explain and discuss.
Response 8: Thank you for your comments. According to your comments, we discussed the great inequity in the number of participants between groups with and without SB episodes in the discussion section as below.
“However, genders were not equally represented in this study. Previous clinic-based and community-based studies have consistently reported a higher prevalence of OSAS in men than in women [33, 34]. Because we compared cases with and without SB episodes among patients with OSAS rather than in healthy controls, this explains why there were more men than women in the composition of patients. However, we found that SB episodes were not associated with gender. Therefore, we postulate that the small number of female subjects in this study was associated with the effect of OSAS rather than with SB episodes.”
Point 9. Table 1: explain the following:
Are these the mean values presented in Table 1 and Table 2? If so, provide such info e.g. Mean (SD).
Response 9: According to your comments, we modified the table 1 and table 2. The mean value provided Mean (SD) is presented in table 1 and table 2.
Point 10. History of SB” meaning?
Response 10: The history of SB stands for self-reported SB. To reduce the potential for misunderstanding, the history of SB was switched to self-reported SB. Self-reported SB is determined based on questionnaires whether the patients had been told or noticed by themselves that they grind their teeth or clench their jaws when they are asleep.
Point 11.“High blood pressure” define and explain whether patients (+) were taking medications or not? Same with DM.
Response 11: Hypertension and diabetes mellitus were determined based on patients’ records at hospital visits, considering factors such as taking drugs prescribed for these conditions. “Hypertension (±) or diabetes mellitus (±) means that the patient was not or was diagnosed and taking medications prescribed for these conditions, respectively.” We added these sentences to the table of the manuscript.
Point 12. Terms such as “supine positional sleep time” or “nonsupine positional sleep” should be verified and corrected.
Response 12: Your comment verified and corrected the terms of supine positional sleep time or non-supine positional sleep, such as “supine positional sleep time” is switched to “sleep time in the supine position.” and “nonsupine positional sleep” is changed to “sleep in the nonsupine position.”
Reference 1. Lisa M Walter et al. Back to sleep or not: the effect of the supine position on pediatric OSA: Sleeping position in children with OSA. Sleep Med. 2017 Sep;37:151-159. doi: 10.1016/j.sleep.2017.06.014. Epub 2017 Jul 3.
☞ Results: All children spent significantly more sleep time in the supine position than in any other position
Reference 2. Millene R. et al. Supine sleep and positional sleep apnea after acute ischemic stroke and intracerebral haemorrhage. Clinics (Sao Paulo). 2012 Dec; 67(12): 1357–1360.
doi: 10.6061/clinics/2012(12)02, PMCID: PMC3521795, PMID: 23295586
☞ A significant correlation was observed between the percent of sleep time in the supine position and the NIHSS (rs = 0.5; p<0.001).
Point 13: Lines 133-134: revise giving more info on the differences between the results obtained for patients with OSAS, with and without SB episodes. Move here the info from lines 170-178 (from Discussion), underline the significant differences.
Response 13: We appreciate your significant comment on the sentences in the results section. We kindly confirmed sentences 170-178 of the discussion part and agreed that they fit in the result section. We rearranged according to the segment from the discussion to the result part to give more information on the differences between the results obtained for patients with OSAS, with and without SB episodes.
Point 14: Line 158: please add “… in patients with OSAS” at the end of the sentence.
Response 14: According to your comment, we added the phrase “… in patients with OSAS” at the end of the sentence of 158 lines.
Point 15: Line 165: revise “…assess the association”
Response 15: According to your comment, we revised the sentence as below.
“The criteria used to assess the association between SB and OSAS have varied between studies, and the results are controversial.”
☞ The criteria used to investigate the relationship between SB and OSAS have varied between studies and the results are controversial [11-16].
Point 16: Line 166: revise, “therefore” twice in one line
Response 16: According to your comment, we modified the context as a whole so that twice do not overlap in one line.
Point 17: Discussion is poorly written. First part consists of either the repetition of statements from Introduction or statements (lines 170-178) that rather belong to the Results section not Discussion.
Response 17: Thank you for the valuable comment on the reader’s understanding of the introduction and discussion section. Based on the reviewer’s statement, we rearranged the sentences from the discussion to the introduction part and avoided the same or similar sentences in the introduction and the discussion. And we modified our introduction and discussion section to help the reader’s understanding. We rearranged the statements (line 170-178) from the discussion part to the result part to give more information on the differences between the results obtained for patients with OSAS, with and without SB episodes.
Point 18: Line 178-180: the statement needs revision as to the causation: low AHI causes SB episodes or SB episodes cause low AHI? In patients with obstructive sleep apnea without SB episodes, severe apnea (high AHI) was observed in both supine and non-supine position during sleep and, in both sleeping positions, AHI was higher than in patients with SB episodes. Please discuss it further.
Response 18: Following your comment, we have added a discussion about causation as follows.
“The AHI in patients without SB episodes was higher than in such patients without SB episodes in both the supine and nonsupine positions during sleep in this study. We found that low AHI was more likely to be linked to SB episodes in OSAS, similar to previous reports [11, 14]. One of the hypotheses relating to SB episodes is that they might protect against OSAS by protruding the mandible and restoring airway patency [14, 22]. However, this mechanism may not be adequate to prevent the airway from collapsing in cases of severe OSAS [14]. Higher oxygen saturation in the study patients was more likely to cause SB episodes in this study. We postulate that when the AHI is high or when the oxygen saturation level is low in patients with OSAS, the mechanisms involved in opening the mouth to breathe are facilitated preferentially over those used to close the mouth for SB episodes, to resolve or compensate for apnea or hypopnea, and to improve oxygen supply. Other mechanisms might be involved in a high AHI, such as excessive respiratory efforts and increased respiratory rate [14]. Therefore, we suggest that a low AHI might induce SB episodes to restore the airway and conversely that a high AHI might reduce or prevent them. However, further studies are needed to verify our hypothesis and determine how high an AHI is associated with reducing or preventing SB episodes.“
Point 19: Line 186: not entirely true as most SB episodes occur in the stages N1 and N2 of sleep, called non-rapid eye movement (non-REM) sleep, what authors mentioned in lines 194-195.
Response 19: Considering your comments and our references, we have revised the mentioned sentence as follows.
“SB episodes can occur during any sleep stage, such as in NREM-like stages N1 and N2 and during REM sleep, or during arousal [26, 27]. In this study, a more prolonged N2 sleep phase was linked to more SB episodes.”
Reference 26. Lavigne GJ, Kato T, Kolta A, Sessle BJ. Neurobiological mechanisms involved in sleep bruxism. Crit Rev Oral Biol Med. 2003;14(1):30-46.
Reference 27. Yap AU, Chua AP. Sleep bruxism: Current knowledge and contemporary management. Journal of conservative dentistry : JCD. 2016;19(5):383-9.
Point 20: Lines 198-199: revise the statement as in this study the age of participants is not given nor genders are equally represented.
Response 20: As mentioned in “Response 7”, descriptions of the subject’s age have been added, and analyzes are provided in the text. Reflecting on your comment and this, the sentences and contents have been modified as follows.
“In our study, however, SB episodes in patients with OSAS were not associated with the patients’ characteristics, including age, gender, or the presence of hypertension or diabetes mellitus. However, genders were not equally represented in this study. Previous clinic-based and community-based studies have consistently reported a higher prevalence of OSAS in men than in women [33, 34]. Because we compared cases with and without SB episodes among patients with OSAS rather than in healthy controls, this explains why there were more men than women in the composition of patients. However, we found that SB episodes were not associated with gender. Therefore, we postulate that the small number of female subjects in this study was associated with the effect of OSAS rather than with SB episodes.”
Point 21: Line 210: again define “patients had history of SB”
Response 21: As mentioned in “Response 10”, the “history of SB” has been changed to “self-reported SB.”
Point 22: Line 217: there are portable devices to measure SB: https://pubmed.ncbi.nlm.nih.gov/31261634/ and https://pubmed.ncbi.nlm.nih.gov/25040303/
Response 22: Based on the reviewer’s comment, we carefully reviewed the suggested two literature and added our manuscripts as references.
“ Easy and comfortable approaches for audio/video recordings, such as smartphone applications [36], along with the use of portable PSG [37, 38], should be developed.”
Reference 37. Manfredini, D.; Ahlberg, J.; Castroflorio, T.; Poggio, C.E.; Guarda-Nardini, L.; Lobbezoo, F. Diagnostic accuracy of portable instrumental devices to measure sleep bruxism: a systematic literature review of polysomnographic studies. J Oral Rehabil 2014, 41, 836-842, doi:10.1111/joor.12207.
Reference 38. Saczuk, K.; Lapinska, B.; Wilmont, P.; Pawlak, L.; Lukomska-Szymanska, M. The Bruxoff Device as a Screening Method for Sleep Bruxism in Dental Practice. Journal of clinical medicine 2019, 8, doi:10.3390/jcm8070930.

Round 2
Reviewer 2 Report
Thank you for revised manuscript. The authors significantly improved the manuscript.
Few points:
Explain abbreviations at their first use
Line 32” add “muscle” in the sentence as in “muscle activity”
Author Response
Thank you for your constructive comments.
Point 1. Explain abbreviations at their first use
Response: Thanks for the comment on the overall form for our manuscript. As the reviewer mentioned, we checked the whole document and explained the first abbreviation used.
Point 2. Line 32” add “muscle” in the sentence as in “muscle activity”
Response: According to your comment, we added muscle to the sentence, as shown below.
“Sleep bruxism (SB) is a masticatory muscle activity during sleep that is characterized as rhythmic (phasic) or nonrhythmic (tonic) and is not a movement disorder or a sleep disorder in otherwise healthy individuals [3].”
